# Cross-Linked Ionic Liquid Polymer for the Effective Removal of Ionic Dyes from Aqueous Systems: Investigation of Kinetics and Adsorption Isotherms

**DOI:** 10.3390/molecules27227775

**Published:** 2022-11-11

**Authors:** A. Vijaya Bhaskar Reddy, Rehan Rafiq, Aqeel Ahmad, Abdulhalim Shah Maulud, Muhammad Moniruzzaman

**Affiliations:** 1Department of Chemistry, Atria Institute of Technology, Bengaluru 560024, Karnataka, India; 2Department of Chemical Engineering, Universiti Teknologi PETRONAS, Seri Iskandar 32610, Perak, Malaysia; 3Centre of Research in Ionic Liquids (CORIL), Institute of Contaminant Management (ICM), Universiti Teknologi PETRONAS, Seri Iskandar 32610, Perak, Malaysia

**Keywords:** ionic liquids, adsorption, reusability, dye substances, Langmuir isotherm, wastewater

## Abstract

In the current study, we have synthesized an imidazolium based cross-linked polymer, namely, 1-vinyl-3-ethylimidazolium bis(trifluoromethylsulfonyl)imide (poly[veim][Tf_2_N]-TRIM) using trimethylolpropane trimethacrylate as cross linker, and demonstrated its efficiency for the removal of two extensively used ionic dyes—methylene blue and orange-II—from aqueous systems. The detailed characterization of the synthesized poly[veim][Tf_2_N]-TRIM was performed with the help of ^1^H NMR, TGA, FT-IR and FE-SEM analysis. The concentration of dyes in aqueous samples before and after the adsorption process was measured using an UV-vis spectrophotometer. The process parameters were optimised, and highest adsorption was obtained at a solution pH of 7.0, adsorbent dosage of 0.75 g/L, contact time of 7 h and dye concentrations of 100 mg/L and 5.0 mg/L for methylene blue and orange-II, respectively. The adsorption kinetics for orange-II and methylene blue were well described by pseudo-first-order and pseudo–second-order models, respectively. Meanwhile, the process of adsorption was best depicted by Langmuir isotherms for both the dyes. The highest monolayer adsorption capacities for methylene blue and orange-II were found to be 1212 mg/g and 126 mg/g, respectively. Overall, the synthesized cross-linked poly[veim][Tf_2_N]-TRIM effectively removed the selected ionic dyes from aqueous samples and provided >90% of adsorption efficiency after four cycles of adsorption. A possible adsorption mechanism between the synthesised polymeric adsorbent and proposed dyes is presented. It is further suggested that the proposed ionic liquid polymer adsorbent could effectively remove other ionic dyes and pollutants from contaminated aqueous systems.

## 1. Introduction

Contamination of aqueous systems with dye effluents poses a significant threat to the ecosystem and has a detrimental impact on human beings and ecosystem due to their toxicity and carcinogenicity [1,2]. Azo dyes are extensively used synthetic organic dyes and account for 60–70% of the total production of dyes. Amongst the several anionic azo dyes, orange-II (O-II) is the most popular and is typically used in leather dyeing, textiles and furniture industry [3]. For decades, it has been known for its acute toxicity, carcinogenicity and mutagenic nature [4]. Cationic azo dyes are commonly used for the colouring of nylon, silk and woollen clothes owing to their near contact with cloth surfaces. A common example of cationic dyes is methylene blue (MB), which induces a range of health problems such as dysfunctional breathing, fatigue, excessive sweating, mental discomfort and methemoglobinemia when discharged into drinking water sources [5].

Considering the high toxicity of dyes, several treatment processes have been reported for their removal from aqueous systems including chemical oxidation, electrochemical oxidation, precipitation, photochemical degradation, membrane filtration and adsorption [6]. Among the reported technologies, adsorption has been recognized as the most successful process due to its easy separation, greater removal efficiency, affordable cost and good material reusability [7]. Over the years, a large number of adsorbents such as metal oxides, zeolites, natural clays, advanced carbon materials and metal nanoparticles have been successfully employed for the removal of dyes from aqueous systems [8,9,10,11,12,13,14]. However, the heightened awareness of green chemistry principles created the necessity for developing low-cost and green adsorbents that can effectively remove environmental contaminants. Recently, ionic liquids (ILs) have received high attention as alternative solvents in view of their remarkable characteristics including negligible vapour pressure, low volatility, high thermal stability and good reusability. As a result, the applications of ILs have been extensively studied in fuel cells, catalysis, nuclear systems and environmental sustainability in recent years [15,16,17,18,19,20,21]. Although ILs have emerged as effective solvents for the extraction/removal of toxic organic pollutants and dyes from contaminated waters, most ILs are viscous and possess considerable solubility in water [22]. Consequently, the regeneration of ILs for their subsequent utilization is critical and necessitates substantial energy [23]. As a result, the polymerization of ILs and their subsequent utilization in adsorption process has gained significant attention in recent times [24,25,26]. Polymerised ILs (PILs) overcome the limitations of conventional ILs and provide additional designability, mechanical resilience and ionic conductivity. Among the various advantages of PILs for which they have received wide attention is their application in catalysis, reactive materials, adsorption and other energy systems [27,28,29,30,31]. Further, the specific adsorbate and adsorbent (PILs) interactions including electrostatic interactions, van der Waal forces and hydrogen bonding are promising in the adsorption of organic dyes [32]. 

PILs effectively adsorb anionic dyes owing to the electrostatic forces of attraction between dyes and the cationic group of the IL. For example, a polymeric adsorbent prepared from polyvinyl benzyl chloride and 2-methylimidazole has provided a maximum adsorption of 2012 mg/g for Acid Orange-7 anionic dye [33]. In another study, a novel N-methyl pyrrolidinium polymer adsorbent provided maximum monolayer adsorption capacities of 198.4 mg/g, 279.3 mg/g and 316.5 mg/g for Orange-G, Orange-II and Sunset Yellow FCF, respectively [34]. Most recently, an imidazolium based PIL was synthesized by surfactant-free IL microemulsion, and was evaluated for its adsorption capacity in the removal of methyl orange and disperse red from contaminated sites. The highest adsorption efficiencies were determined to be 187 mg/g and 1080 mg/g for methyl orange and disperse red, respectively [35]. In addition, a superabsorbent polymer (SAP) prepared from starch phosphate carbamate (SPC) and acrylamide (AM) has shown an adsorption capacity of 62.52 mg/g for the removal of methylene blue [36]. In addition, several studies demonstrated the ability of microporous polymeric adsorbents for the removal of various dyes and metal ions from aqueous systems [37,38,39]. 

The existing data suggests that many studies have been conducted for the removal of a single category of dyes (either anionic or cationic) using various PILs. By contrast, there are no studies reported for the simultaneous removal of anionic and cationic dyes. Thus, in the present investigation, we have synthesized a cross-linked poly[veim][Tf_2_N]-TRIM adsorbent and demonstrated its adsorption efficiency towards MB and O-II dyes from aqueous systems. The cross-linker trimethylolpropane trimethaacrylate (TRIM) was incorporated into the polymer adsorbent to control the morphology and physical properties of the polymer matrix, to provide additional mechanical stability and to stabilize the binding sites of the polymer matrix [40]. In fact, TRIM is a superior crosslinker that provides excellent binding capacity to the polymer matrix through its vinyl groups. Several studies have already confirmed that the incorporation of TRIM during the synthesis of polymer adsorbents significantly enhanced their adsorption capacity for various analytes and pollutants [41,42,43]. For instance, a study concluded that microparticles synthesised using the TRIM cross-linker have shown higher adsorption properties towards 5-fluorouracil than those synthesized without a cross-linker. Similarly, a few TRIM based molecularly imprinted polymers (MIPs) confirmed the usefulness of cross-linkers in adsorption studies. Further, a recent study reported that the usage of an appropriate amount of cross-linker is essential to obtain a porous structure with high mechanical strength and to keep the binding sites [44]. Furthermore, our previous study confirmed the surface area and pore volume of poly[veim][Tf_2_N]-TRIM through N_2_ adsorption desorption measurements, which were found to be 27 m^2^/g and 0.14 cm^3^/g respectively [45]. Although the values are relatively lower than the other micelle incorporated poly[veim][Tf_2_N]+SMI materials, the results confirmed the porous structure and availability of binding sites in poly[veim][Tf_2_N]-TRIM for the selected dyes. In addition, the synthesized PIL have certain advantages over other reported adsorbents including its high adsorption capacity, simple process optimization and process adaptability. The dye removal occurred at mild operating conditions and the process does not produce any toxic by-products. The impact of various process parameters including the initial dye concentration, adsorbent dosage, contact time, solution pH and temperature on adsorption efficiency of the proposed polymer adsorbent was examined to determine the optimum conditions that produce maximum adsorption. Moreover, the kinetic parameters and adsorption isotherms were assessed comprehensively.

## 2. Experimental 

### 2.1. Chemicals and Reagents

Bromoethane and 1-vinyl imidazole were purchased in their highest grades (≥99%) from Merck (Darmstadt, Germany). Lithium bis(trifluoromethyl)sulphonamide (LiTf_2_N, ≥99%), 2,2′-azobis(2-methylpropionitrile), and trimethylolpropane trimethacrylate were purchased from Sigma Aldrich (St. Louis, MO, USA). Next, the dye standards—namely, methylene blue (>99%) and orange-II (>99%), and ethanol, ethyl acetate and acetone were purchased from Fisher Scientific (Loughborough, UK) in AR Grade. A highly precise Sartorius^®^-CP124S competence laboratory balance was used to weigh the standards and to prepare standard solutions. The synthesized ILs were tightly closed and stored in a desiccator to prevent hydration. Millipore water was employed during the preparation of standard and stock solutions.

### 2.2. [veim][Br] and [veim][Tf_2_N] Monomer Synthesis

The detailed preparation of [veim][Br] and [veim][Tf_2_N] monomers is presented in our previous study [46]. Briefly, 1-vinyl-3-ethylimidazolium bromide [veim][Br] was synthesized by slowly adding 0.3 mol^−1^ (33 g) bromoethane to the round bottom flask that contained 0.3 mol^−1^ (28 g) 1-vinylimidazole and stirred vigorously for 2 h at 40 °C. At completion of the reaction, the [veim][Br] monomer was produced as white precipitate, which was dissolved in ethyl acetate to complete the precipitation and to isolate the unreacted materials. The washing procedure was repeated several times with ethyl acetate and finally the monomer was dried under reduced pressure using a rotary evaporator (rotavapor) for about 6 h. The resulting [veim][Br] monomer was analysed using ^1^H NMR for the confirmation of its chemical structure.

Similarly, the IL monomer 1-viny-3-ethylimidazolium bis(trifluoromethyl-sulfonyl)imide [veim][Tf_2_N] was synthesised by a simple ion exchange process between [veim][Br] and lithium bis(trifluoromethyl)sulphonamide (Li[Tf_2_N]). A slight excess of anionic salt was added at a molar ratio of 1:1.1 to ensure the total ion exchange and the mixture was continuously stirred for 8 h until the formation of two clear separate layers of [veim][Tf_2_N] and LiBr. After the formation of clear layers, the [veim][Tf_2_N] obtained in the bottom phase was isolated and rinsed with distilled water to eliminate the unreacted reactants and then dried on a rotavapor to remove the moisture and other residual solvents. The formation of [veim][Tf_2_N] was ensured through ^1^H NMR analysis.

### 2.3. Synthesis of Cross-Linked poly[veim][Br] and poly[veim][Tf_2_N]

The cross-linked poly[veim][Br]-TRIM and poly[veim][Tf_2_N]-TRIM materials were synthesized using free radical polymerization. To synthesize the poly[veim][Br]-TRIM, equal amounts (*w*/*w*) of [veim][Br] (1.0 g) and ethanol (1.0 g) were taken into the flask and used along with nitrogen to purge the mixture for 15 min. After purging, the reaction mixture, 20 mg of free radical initiator (AIBN) and 50 mg of cross-linker (TRIM) were added as cross-linker and again used to purge the reaction mixture for 10 min under nitrogen; the reaction mixture was subsequently stirred in oil bath at 65 °C for approximately 30 min until the formation of an elastic gel-type polymeric material. The resulted polymeric gel was diluted with a small fraction of reaction solvent and then acetonitrile was added to precipitate the poly[veim][Br]-TRIM. The resultant product was then rinsed, dried and stored in a desiccator.

Similarly, the synthesis of poly[veim][Tf_2_N]-TRIM was conducted by adding equal amounts of [veim][Tf_2_N] (1.0 g, 0.003 mol) and ethanol (1.0 g, 0.002 mol) to a flask and purging the mixture with nitrogen for 30 min. Afterwards, the solution was enriched with 20 mg of AIBN (0.12 mol) and 50 mg of TRIM (0.15 mmol) and again purged with nitrogen for 10 min. The reaction mixture was then stirred at 65 °C for about 30 min until it formed a polymeric gel. After the polymerization was completed, a small fraction of reaction solvent was added to dilute the resultant product, and subsequently poly[veim][Tf_2_N]−TRIM was precipitated by pouring into water. The resultant polymer was initially dried and preserved in a desiccator until further use. The schematic representation for the formation cross-linked PIL-TRIM is presented in Figure 1.

### 2.4. Characterization of Synthesized IL Monomer and Polymers

The structural confirmation of synthesised IL monomers was ascertained using INOVA 500 MHz NMR spectrometer. For the ^1^H NMR analysis, the IL monomers were dissolved in deuterated methanol, mixed, filtered and clear samples were poured into NMR tubes. Next, the FT-IR spectra of prepared IL polymers were scanned between 4000–600 cm^−1^ using Shimadzu IR-Tracer-100 fitted with a diamond ATR module [38]. Thereafter, the surface morphology of synthesized PIL materials was characterized using FE-SEM (S-4300, Hitachi), before which the samples were precisely ground to form a thin layer of gold coating using the evaporator. The thermal characterization of synthesised IL polymers was conducted using thermogravimetric analysis (TGA/DTG, Perkin Elmer Simultaneous Analyzer STA 6000). The thermograms were recorded at a temperature range between 50–700°C with 10 °C/min heat increments under an inert N_2_ atmosphere. The measurements were taken with a precision of ±0.1% weight.

### 2.5. Dye Adsorption Experiments

Two analytes namely MB (cationic dye) and O-II (anionic dye) were selected as model analytes to determine the adsorption efficiency of synthesised PIL adsorbents. Initially, the individual standard stock solutions of dyes were prepared by adding an accurately weighed quantity of dyes to distilled water. From the above standard stock solutions, six calibration standards of each dye were prepared through serial dilution. An amount of 0.1 M NaOH and 0.1 M HCl solutions were used to adjust the pH of the samples. The effect of adsorbent dosage on overall adsorption was assessed by altering the initial adsorbent concentration in the aqueous dye solution. The stirring speed of the test solution was maintained at 100 rpm for a predetermined time at ambient temperature. The standard adsorption experiments were performed with 100 mL of aqueous solution containing 50 mg/L of each dye at 25 °C for 7 h. After the reaction time was complete, the test samples were centrifuged and the aqueous phase subsequently analysed using UV-vis spectrophotometer to determine the concentration of unextracted dyes. The extraction efficiency (%E) and adsorption capacity Q (mg/g) of synthesized PIL-TRIM towards each dye was assessed using the following Equations (1) and (2):(1)$$E\%=\frac{C_{0}-C_{e}}{C_{0}}\times 100$$
(2)$$Q_{e}=\left(C_{0}-C_{e}\right)\left(\frac{V}{m}\right)$$
where, C_0_ and C_e_ are initial and equilibrium dye concentrations (mg/L), respectively; V is test sample volume (L); and m is adsorbent mass (g).

### 2.6. Desorption

To determine the reusability of prepared PIL adsorbents, desorption studies were performed using a mixture of choline chloride in water and ethanol (5.0 wt.% in 1:1 *v*/*v*). For this, 50 mg of post-adsorption material was taken and soaked in 5.0 mL of the aforementioned solution mixture and slowly stirred for about 30 min at room temperature. Later, the polymeric adsorbents were filtered, washed and dried at 50 °C under vacuum. The processed adsorbent was re-examined to determine its performance efficiency towards the removal of the same dyes. The desorption experiments were repeated for four cycles to compare the removal efficiency of the PIL-TRIM adsorbent and its lifespan.

## 3. Results and Discussions

### 3.1. Synthesis and Characterization of PIL-TRIM Adsorbent

Prior to the synthesis of PILs, the prepared monomers [veim][Br] and [veim][Tf_2_N] were characterized for their structural confirmation using ^1^H NMR. The results confirmed the structure and purity of synthesized IL monomers. The ^1^H NMR data of [veim][Br] and [veim][Tf_2_N] confirmed the successful synthesis of IL monomers evidenced through the total number of hydrogens in each monomer.

^1^H NMR data of [veim][Br] (500 MHz; MeOD): δ 9.50 (s, 1H, N-C*H*-N), 8.20 (s, 1H, N-C*H*-CH-N), 7.85 (s, 1H, N-CH-C*H*-N), 7.30 (dd, 1H, N-C*H*-CH_2_); 5.95 (dd, 1H, N-CH-C*H*_2_), 5.32 (dd, 2H, N-CH-C*H*_2_); 4.22 (q, 2H, N-C*H*_2_-CH_3_); 1.55 (t, 3H, CH_2_-C*H*_3_). 

^1^H NMR data of [veim][Tf_2_N] (500 MHz; MeOD): δ 9.20 (s, 1H, N-C*H*-N); 7.95 (s, 1H; N-C*H*-CH-N); 7.75 (s, 1H, N-CH-C*H*-N); 7.25 (dd, 1H, N-CH-CH_2_); 5.90 (dd, 1H, N-CH-C*H*_2_), 5.45 (dd, 2H, N-CH-C*H*_2_); 4.30 (q, 2H, N-C*H*_2_-CH_3_); 1.60 (t, 3H, CH_2_-C*H*_3_).

The synthetic route affirmed for the preparation of poly[veim][Br]-TRIM and poly[veim][Tf_2_N]-TRIM is presented in Figure 2. The formation of cross-linking in imidazolium backbone was intended to enhance the physiochemical properties of the resultant polymer and subsequently improved the sorption efficiency of PIL-TRIM. In addition, the aromatic rings on imidazolium improves the π–π interactions between adsorbent and adsorbate molecules, and subsequently improves the overall adsorption potential of polymer [34,39].

The FT-IR scanning of poly[veim][Br]-TRIM and poly[veim][Tf_2_N]-TRIM confirmed the formation of the respective polymers and the incorporation of TRIM during polymerisation. The corresponding FT-IR spectra of poly[veim][Br]-TRIM and poly[veim][Tf_2_N]-TRIM are presented in Figure 3. The three peaks appeared between 2920–3100 cm^−1^ in both poly[veim][Br]-TRIM and poly[veim][Tf_2_N]-TRIM spectra representing the alkene C–H (C=C–H; sp^2^) stretch of the imidazolium ring. Further, the peaks that appeared at 1675 cm^−1^ and 1550 cm^−1^ correspond to the C=C and C=N stretch of the imidazolium ring, respectively [33,45]. Next, a couple of peaks detected at 1485 cm^−1^ and 1160 cm^−1^ correspond to C–H bending and C–N bending of the imidazolium ring, respectively. Additionally, the two peaks that appeared at 1355 cm^−1^ and 1055 cm^−1^ only in poly[(veim)(Tf_2_N)-TRIM] are characteristic peaks of SO_2_ symmetric and asymmetric bonding on Tf_2_N anions, respectively. At last, the peak at 1730 cm^−1^ in both the polymers is the characteristic peak of trimethylolpropane trimethacrylate (TRIM) corresponding to the C=O group.

The surface morphology of the synthesized polymers viz., poly[veim][Br]-TRIM and poly[veim][Tf_2_N]-TRIM, was examined with the aid of FE-SEM. The morphology results helped to assess the porous nature of the polymers based on the particle sizes in frameworks (1.0 μm–200 nm) as manifested in Figure 4. Figure 4a,b depicts that the surface of poly[veim][Tf_2_N]-TRIM is relatively more porous than that of poly[veim][Br]-TRIM. In support of this, EDX analysis of elemental composition confirmed the higher content of carbon, oxygen, and fluorine atoms in poly[veim][Tf_2_N]-TRIM than in poly[veim][Br]-TRIM which could enhance the dye adsorption. Further, the surface morphology of poly[veim][Tf_2_N]-TRIM after MB and O-II adsorption is conveyed in Figure 4c,d, wherein the adsorption of dye molecules onto the adsorbent can be seen clearly while surface roughness of the adsorbent has been altered. Overall, there is a clear difference in surface morphology of poly[veim][Tf_2_N]-TRIM before and after dye adsorption. 

The higher thermal stability of synthesised IL polymers is another prerequisite for their potential application in dyes adsorption. Hence, the thermal stability of poly[veim][Br]-TRIM and poly[veim][Tf_2_N]-TRIM was evaluated through thermogravimetric analysis (TGA) under inert N_2_ atmosphere. The TGA plots presented in Figure 5 provide precise information about the thermal stability of both the polymers. As seen in Figure 5, the poly[veim][Tf_2_N]-TRIM showed higher T_10%_ (Temperature at 10 wt.% loss) and slower decomposition rates compared to poly[veim][Br]-TRIM. From the TGA data, it is perceived that the anion influenced (Tf_2_N>Br) the thermal stability of the polymers, which is in good agreement with reported studies [45,46]. Specifically, the findings revealed that the weight loss of samples began around 50 °C. The cleavage of ethyl group connecting to the imidazolium ring might have contributed to a weight loss of 7 percent. The two-step decomposition process at around 300 °C and 400 °C provisionally confirmed the formation of cross-linked polymeric structures. Finally, at 450 °C, the polymers substantially decomposed because of the breakage of the large polymeric chains of the respective polymers. A good degree of cross-linking in the polymers provided relatively higher thermal stability [35,47]. The higher thermal stability of synthesized PILs demonstrates a higher degree of polymerization and good cross-linking formation among the polymeric chains, and their subsequent ability to apply at high temperatures during the adsorption of dyes. Additionally, the derivative thermogravimetric (DTG) plots in Figure 5 provided a precise and accurate information package regarding the thermal stability of the PILs. The respective peaks indicated that poly[veim][Tf_2_N]-TRIM is more stable than poly[veim][Br]-TRIM.

### 3.2. Study of Adsorption Performance

The adsorption capacity of synthesised PIL materials was examined to assess their adsorption capacity towards the selected dyes. To evaluate the influence of anions attached to the PILs, the two cationic and anionic dyes—respectively, MB and O-II—were selected as representative adsorbates, and the adsorption efficiency of both PIL adsorbents was then examined using a UV-vis spectrophotometer. Figure 6 shows the adsorption performance of poly[veim][Br]-TRIM and poly[veim][Tf_2_N]-TRIM towards the removal of the selected dyes. The initial experiments were conducted with 50 mg/L of dye concentration, 0.5 g/L of adsorbent dosage, solution pH of 6.5 and stirring speed of 100 rpm for about 7 h to ensure complete adsorption.

As shown in Figure 6, the poly[veim][Br]-TRIM adsorbent showed relatively good adsorption efficiency for the anionic dye, i.e., O-II, but its ability towards the adsorption of MB is very poor. This can be explained by the electrostatic repulsions between MB and poly[veim][Br]-TRIM that might have inhibited the adsorption of MB onto the adsorbent, while the SO_3_^-^ functional group of the O-II dye could have favoured the dye adsorption via electrostatic interactions. The presence of Br^¯^ ion in poly[veim][Br]-TRIM enhanced the surface positive charge. Accordingly, the cationic dye (MB) adsorption was reduced, and the anionic dye adsorption was increased. By contrast, in the case of poly[veim][Tf_2_N]-TRIM, the removal of MB and O-II dyes was found to be 98% and 36%, respectively. This is because poly[veim][Tf_2_N]-TRIM was prepared by ion-exchange reaction, in which bulk cation combines effectively with bulk anion in accordance with the hard/soft-acid/base principle (HSAB) [48,49], so a non-homogeneous charge distribution produces considerable potential surface power and leads to the adsorption of both ionic dyes to neutralize some electric charge on the surface via electrostatic interactions. It was reported that imidazolium-based ILs with the Tf_2_N anion showed relatively high extraction efficiency for MB [18]. The difference in adsorption capacity for both individual dyes was because of their dissimilar functional groups. The steric hindrance between the SO_3_^-^ and Tf_2_N anions caused low adsorption efficiency for O-II, but it is evident that the N^+^ atom of the imidazolium ring along with the Tf_2_N anion can interact electrostatically with the SO_3_^−^ group of O-II [34]. Moreover, the surface morphology of poly[veim][Tf_2_N]-TRIM analysed by FE-SEM indicates that the material is exceedingly more porous than the poly[veim][Br]-TRIM, which has showed high adsorption potential for both ionic dyes. These results suggested that the Br^−^ based PIL adsorbent, i.e., poly[veim][Br]-TRIM, is favourable only for the adsorption of anionic dyes, while the Tf_2_N based PIL adsorbent, i.e., poly[veim][Tf_2_N]-TRIM, is capable of adsorbing both cationic and anionic dyes effectively. Therefore, the Tf_2_N anion based PIL i.e., poly[veim][Tf_2_N]-TRIM, was selected for further optimization studies.

#### 3.2.1. Effect of Solution pH

The effect of solution pH on the adsorption of MB and O-II was assessed because the surface charge of adsorbent and degree of ionization of substrate are pH dependent, and they determine the overall adsorption capacity [50,51,52,53]. Adsorption tests were performed taking the dye concentrations as 20 mg/L of O-II and 500 mg/L of MB with respect to 0.5 g/L of adsorbent at varying pH values between 2.0–12.0. As shown in Figure 7A, the MB adsorption onto poly[veim][Tf_2_N]-TRIM was magnified with a rise in solution pH, because the MB adsorption is anticipated by the chemical and electrostatic interactions between the adsorbate and surface of adsorbent molecules. The increase in solution pH enhanced the OH^−^ ion concentration in aqueous solution and the surface of adsorbent underwent deprotonation which intensified the negative charge. Therefore, electrostatic attractions between a positive dye molecule and negative adsorbent surface is increased [54]. Subsequently, the rate of adsorption of MB onto poly[veim][Tf_2_N]-TRIM was raised by increasing the pH of solution. By contrast, in the case of O-II, a reverse trend was observed with respect to solution pH, which is mainly because of electrostatic repulsion between the negatively charged dye molecule and the adsorbent surface charge at higher pH values. The high extraction of O-II at low solution pH was mainly due to the high concentration of proton ions in the adsorbent, which created electrostatic attraction with the dye molecules and increased the extraction efficiency of O-II [55]. The findings revealed that electrostatic forces are crucial and preside over the adsorption of MB and O-II onto poly[veim][Tf_2_N]-TRIM. In view of the above findings, an optimum solution pH of 7.0 was maintained for further optimisation.

#### 3.2.2. Effect of Adsorbent Dosage

To determine the minimum amount of adsorbent that can provide maximum dye removal efficiency, a proper optimization of adsorbent concentration is required. The extraction efficiency of adsorbent was observed by adding various amounts, i.e., 0.1, 0.25, 0.5, 0.75 and 1.0 g/L of poly[veim][Tf_2_N]-TRIM to the aqueous dye solutions. Figure 7B indicates that the dye removal was proportionally increased with adsorbent concentration as the number of adsorption sites for dyes are increased with adsorbent quantity, which is in good concurrence with the published studies [34,56,57]. Approximately 80 percent of dyes from aqueous solution were extracted on to adsorbent in 7 h of contact time with a dosage of 0.75 g/L. However, the adsorption rate was not progressively increased by further increasing the adsorbent dosage beyond 0.75 g/L, which implies that the adsorbent saturation potential might not be achievable. For dye-contaminated waters at high concentration, large surface area and higher functional adsorption sites could be available by adding a sufficient amount of adsorbent [58]. Taking into consideration the dye adsorption capacity and removal efficiency, 0.75 g/L of adsorbent dosage was specified as the optimum value for adsorption tests.

#### 3.2.3. Effect of Temperature

The effect of temperature on the adsorption process was examined through controlled adsorption experiments varying the temperature from 25 °C to 55 °C as depicted in Figure 7C. At higher temperatures, the diffusion of adsorbate molecules over the boundary layers was found to be high, and hence the solution viscosity decreased. Furthermore, the change in temperature affected the adsorbent equilibrium capacity [59]. As delineated in Figure 7C, the rate of removal for the O-II dye increased from 74.5% to 84.6% when the temperature altered from 25 °C to 55 °C; similarly, for MB it increased from 74.3% to 81.8%. The results were comparable with the reported studies, meaning that a rise in temperature leads to a higher adsorption capacity [60,61]. As the higher temperature is beneficial to the adsorption of both ionic dyes by poly[veim][Tf_2_N]-TRIM, this indicates that dye adsorption is an endothermic operation.

#### 3.2.4. Effect of Contact Time

The influence of contact time on adsorption efficiency of poly[veim][Tf_2_N]-TRIM for the selected dyes was examined between 0–30 h. Figure 7D demonstrates the impact of contact time on the rate of adsorption at an initial concentration of 20 mg/L for O-II and 500 mg/L for MB. The test solutions were kept with the adsorbents for up to 30 h in order to assess the adsorption potential at equilibrium. It can be seen that the dyes’ adsorption was found to be rapid during the initial hours as the availability of adsorption sites are greater in number and the adsorbent contains the porous poly[veim][Tf_2_N]-TRIM framework, resulting in a powerful binding and diffusion of the dyes on the surface of the adsorbent. Later, the adsorption rate gradually decreased with the decrease in total number of vacant sites on adsorbent surface. The slow rate of adsorption especially towards the end of the experiments shows that the adsorbent surface may have a monolayer formation of MB [62,63]. This might be attributed to the shortage of accessible active sites necessary for more adsorption when the equilibrium has been reached. It is worth mentioning that O-II needed a lengthier time to achieve equilibrium. For consistency, the optimum equilibrium time for both the dyes was taken as 7 h which is more rapid than other reported Tf_2_N anion based IL polymers for the removal of O-II by adsorption [34]. At the completion of the adsorption process, MB showed relatively higher adsorption than O-II (Figure 7D).

#### 3.2.5. Effect of Dye Concentration

The impact of dye concentration on the extraction/adsorption process was examined in this study. The initial dye concentration is a leading force to diminish the mass transfer resistance of dyes between the aqueous phase and polymer adsorbent. Figure 7E,F show the adsorption behaviour of poly[veim][Tf_2_N]-TRIM with respect to dye concentration. The adsorption of dyes gradually declined with the increase in dye concentration, because the binding sites on the adsorbent reduce with increase in dye concentration. In particular, the efficiency of dye removal decreased from 97% to 35%, and 87% to 25% for MB and O-II, respectively, with an increase in dye concentration from 100 to 1000 mg/L and 5 to 200 mg/L, respectively. The synthesised poly[veim][Tf_2_N]-TRIM adsorbent was more effective for the removal of MB at all concentrations compared to O-II. Conversely, the quantity of adsorbed dye per adsorbent mass Q_e_ (mg/g) enhanced relative to rise in dye concentration as seen in Figure 8, which is further explored in an adsorption isotherm study. These findings were well matched with the previous reports [64]. The rise in Q_e_ with respect to initial dye concentration is attributed to the higher dye gradient that served as the driving force for the mass transfer operation, permitting the introduction of additional dye molecules to facilitate the adsorption on the adsorbent surface [34,65].

### 3.3. Adsorption Kinetic Study

Adsorption kinetics predict the pace at which an analyte/pollutant is extracted from aquatic solutions, and also provide useful information regarding the adsorption mechanism. The adsorption kinetics of MB and O-II dyes were best fitted into pseudo-first-order and pseudo-second-order kinetic models, respectively. Figure 9 displays the experimentally obtained adsorption results and their kinetic plots, which are non-linearly fitted to MB and O-II by pseudo-first-order and pseudo-second-order kinetic models, respectively. The determined kinetic parameters are presented in Table 1.

During the adsorption of MB onto poly[veim][Tf_2_N]-TRIM, the coefficient of determination (R^2^) in the pseudo-second-order model was found to be greater than in the pseudo-first-order model since the difference between the theoretical and experimental adsorption capacities is less. Thus, MB adsorption by poly[veim][Tf_2_N]-TRIM is better fitted into the pseudo-second-order kinetic model. The chemical adsorption process is a key control step that determines the adsorption rate and the chemical relationship occurring between MB and the polymer adsorbent [34,66,67]. The EDX analysis further confirmed the decrease in nitrogen content due to anion exchange between MB and poly[veim][Tf_2_N]-TRIM counter ions. As a result, a strong chemisorption interplayed between adsorbent and MB, due to which MB attached to the backbone of the cross-linked polymer. By contrast, in the adsorption of O-II onto poly[veim][Tf_2_N]-TRIM, the coefficient of determination (R^2^) obtained in the pseudo-first-order kinetic model was found larger compared to the pseudo-second-order kinetic model. Moreover, the adsorption data of the pseudo-first-order kinetic model is closer to the experimental data, which is supported by a low ARE error value. These results demonstrated that O-II dye adsorption onto poly[veim][Tf_2_N]-TRIM is best described by the pseudo-first-order kinetic model, indicating that the diffusion process is a rate control step [35].

### 3.4. Adsorption Isotherms

Freundlich and Langmuir adsorption models were selected to fit the O-II and MB adsorption isotherms and to examine the mechanisms involved between the dye molecules and poly[veim][Tf_2_N]-TRIM. The curve fittings of both the models are presented in Figure 9. The correlation coefficients (R^2^), average relative error (ARE), and determined Langmuir and Freundlich isotherm constants are shown in Table 2. The data presented in Table 2 reveal that ARE values are minimum and R^2^ values for MB and O-II dyes adsorption by poly[veim][Tf_2_N]-TRIM approached unity in the Langmuir model; whereas in the Freundlich adsorption isotherm model, ARE values are relatively high and R^2^ values deviate from unity. Therefore, it is confirmed that the experimental results are best fitted into a Langmuir adsorption model rather than a Freundlich model, suggesting a monolayer adsorption [68]. The maximum monolayer adsorption capacity of MB and O-II reached Q_m_ values of 1212 mg/g and 126 mg/g, respectively. The anionic Tf_2_N in the imidazolium backbone and cavities created by the cross-linked structure are key characteristics for higher adsorption of MB. The lower magnitude of Langmuir constant (K_L_) indicates the lower affinity between adsorbed molecules and porous adsorbent surface, which means the adsorbent surface is not entirely occupied by adsorbed molecules and additional adsorption may take place with a rise in concentration until it achieves equilibrium. Next, the separation factor (R_L_) with reference to the Langmuir isotherm model was another significant parameter calculated for both MB and O-II dyes at varying concentrations and the results are presented in Figure 9. In both instances, the R_L_ value relative to concentration was seen to be below unity and above zero, an indicator of their favourable adsorption nature, given that the R_L_ value is a good sign of the adsorption method.

The maximum adsorption (Q_m_) capacity of poly[veim][Tf_2_N]−TRIM adsorbent is compared with the efficiency of other reported adsorbents for the removal of the same dyes from aqueous solutions (Table 3). The adsorption capacity of poly[veim][Tf_2_N]−TRIM towards MB and O-II removal is significantly greater and comparable to that of reported adsorbents in the literature. In particular, the adsorption capacity of poly[veim][Tf_2_N]−TRIM towards MB was higher than all other reported adsorbent materials. In the case of O-II adsorption, the proposed adsorbent is relatively effective for treating the mixture of dyes. The high adsorption capacity of poly[veim][Tf_2_N]−TRIM towards ionic dyes might have been induced by the greater surface area, unique surface morphology, strong hydrogen bonding and good electrostatic interaction of dye with selected adsorbent. Consequently, the adsorbent suggested for MB and O-II in this analysis may be viewed as a promising substitute for ionic dye removal and their separation from aqueous solution.

### 3.5. Reusability Performance

The development of a stable and robust adsorbent that can effectively perform multiple adsorption–desorption cycles with respect to the target substrate is a positive feature since the process cost depends on its reusability [73]. The adsorbent was accessibly separated through filtration after each cycle of dye adsorption. The recycling of spent adsorbent was carried out by immersing it in a mixture of choline chloride–aqueous ethanol solution chosen for its low-cost and environmentally benign nature. As seen in Figure 10, after the 1st cycle, the adsorption efficiency decreased very slightly, but the recycled adsorbent efficiency was sustained above 90% in the following cycles. The instant decrease in adsorption efficiency at the 4th step indicates that even after desorption, most active areas are over-saturated. In addition, the negligible weight loss of adsorbent after the post-adsorption process suggested the stability of the adsorbent for multiple cycles. The findings showed long-term reliability in adsorption and desorption processes of the poly[veim][Tf_2_N]−TRIM, while a slight drop in dye adsorption occurred due to a small insufficiency of desorption.

## 4. Adsorption Mechanism

A proper understanding of adsorption mechanism is very important to assess the adsorption efficiency of the proposed adsorbent. The structures of the dyes and poly[veim][Tf_2_N]-TRIM indicates a charge induced adsorption mechanism between the dye molecules and the ionic liquid polymer (Figure 11). The synergistic effects together with π–π interactions, electrostatic interactions and hydrophobic interactions are attributed to the adsorption of ionic dyes with the polymeric adsorbent, with electrostatic interaction having the major effect [24,52]. In our case, chemical adsorption constitutes the primary adsorption mechanism as demonstrated by adsorbent/dye characterization, adsorption kinetics and isotherms.

The first step towards a potential mechanism of adsorption is to closely investigate both the adsorbent and the adsorbate’s chemical structures. The positive N^+^ atom of the imidazolium ring and negative NTf_2_^−^ anion of poly[veim][Tf_2_N]−TRIM will impair with adsorbates through electrostatic attractions. Hence, the SO_3_^−^ group of the O-II dye interacts with the N^+^ atom of the imidazolium ring, and the N^+^ atom of the MB dye attaches to the NTf_2_^−^ anion electrostatically. There are hydrogen bonds between the ionic dyes and the polymer groups comprising hydroxyl and nitrogen. In addition, aromatic moieties present in adsorbent and adsorbates could lead to enhanced adsorption by π–π interactions among phenyl rings of the dye molecules and polymer structure. Therefore, superior electrostatic interactions, improved hydrogen bonding and π–π interactions play a critical role in improving the adsorption capacity of ionic dyes onto the poly[veim][Tf_2_N]-TRIM polymer.

## 5. Conclusions

In this paper, a cross linked poly[veim][Tf_2_N]-TRIM adsorbent was synthesized via radical polymerization and used for the removal of two ionic dyes, namely, MB and O-II. The prepared adsorbent was found effective towards the removal of both MB and O-II dyes under optimized conditions. The pseudo-first-order kinetic model explained the adsorption data of O-II; and we observed the reverse behaviour for MB adsorption for which the data is well fitted into a pseudo-second-order kinetic model. Further, the Freundlich and Langmuir adsorption models were used to fit the experimental data. In particular, the adsorption data of O-II and MB were best fitted with the Langmuir adsorption model, and the maximum adsorption capacities obtained were 126 mg/g and 1212 mg/g, respectively. Therefore, the proposed adsorbent could be an effective alternative for the simultaneous removal of selected anionic and cationic dyes from contaminated aqueous systems. However, our future studies will examine the impact of a cross linker on the porosity of poly[veim][Tf_2_N], and the possible ion-exchange process between a synthesized IL polymer and analytes/regeneration solvents.

## Figures and Tables

**Figure 1 molecules-27-07775-f001:**
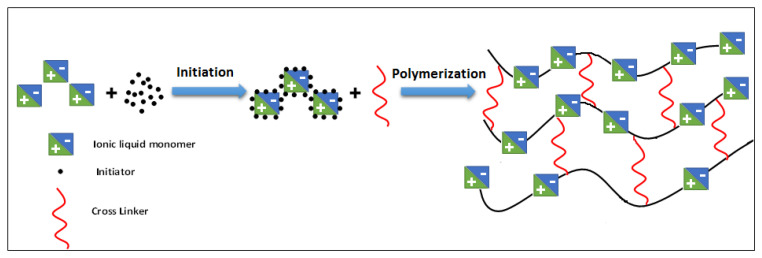
Schematic representation for the formation of the cross-linked poly[veim][Tf_2_N]−TRIM structure.

**Figure 2 molecules-27-07775-f002:**
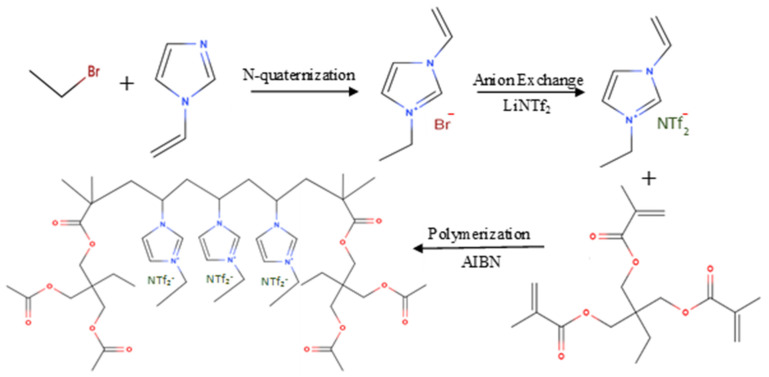
Schematic representation for the synthesis of poly[(veim)(Tf_2_N)−TRIM].

**Figure 3 molecules-27-07775-f003:**
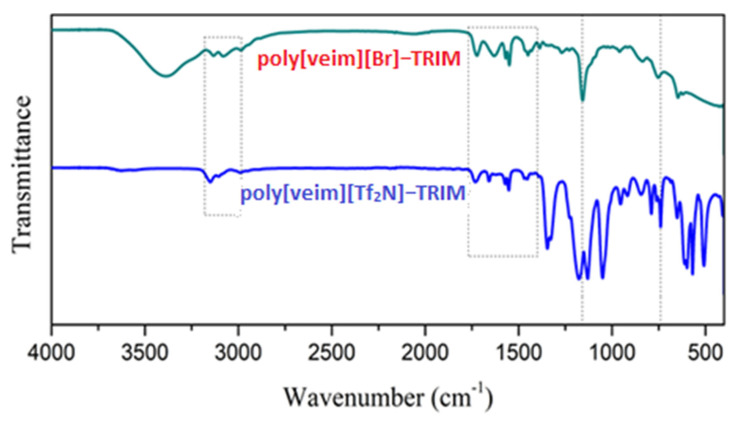
FT−IR spectra of poly[veim][Br]−TRIM and poly[veim][Tf_2_N]−TRIM adsorbents.

**Figure 4 molecules-27-07775-f004:**
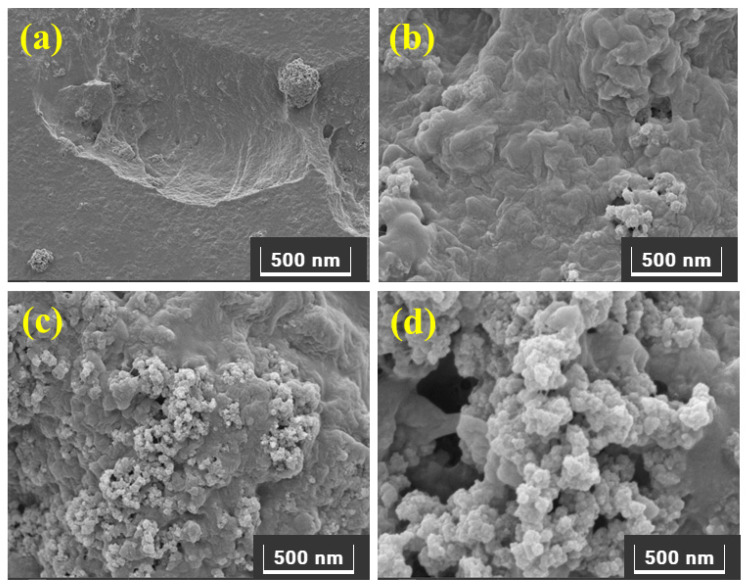
Surface morphology of (**a**) poly[veim][Br]-TRIM, (**b**) poly[veim][Tf_2_N]-TRIM, (**c**) poly[veim][Tf_2_N]-TRIM after MB, and (**d**) poly[veim][Tf_2_N]-TRIM after O-II adsorption.

**Figure 5 molecules-27-07775-f005:**
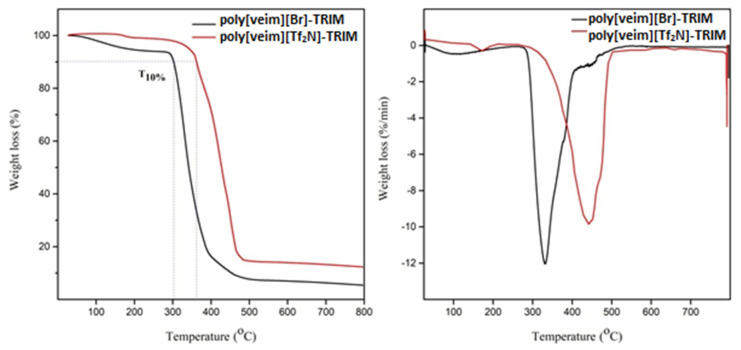
TGA and DTG plots for poly[veim][Br]-TRIM and poly[veim][Tf_2_N]-TRIM.

**Figure 6 molecules-27-07775-f006:**
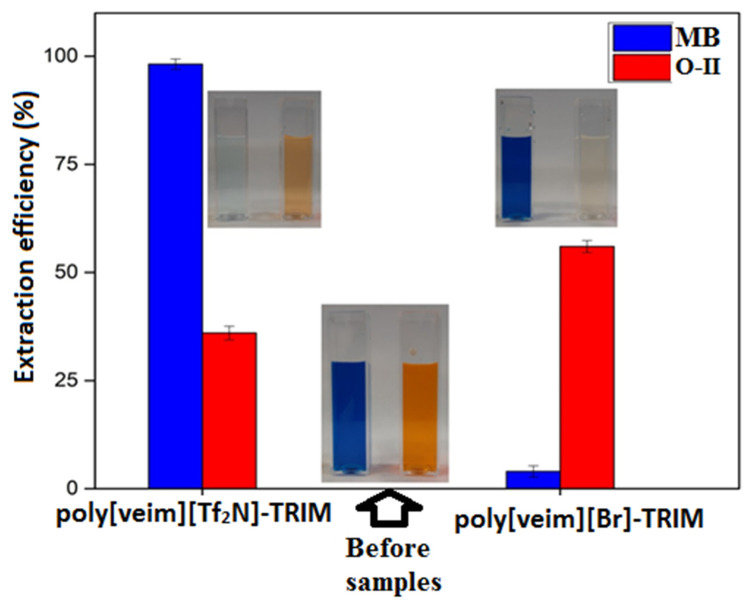
Dye extraction efficiency with different adsorbents (C_0_ = 50 mg/L, m = 0.5 g/L).

**Figure 7 molecules-27-07775-f007:**
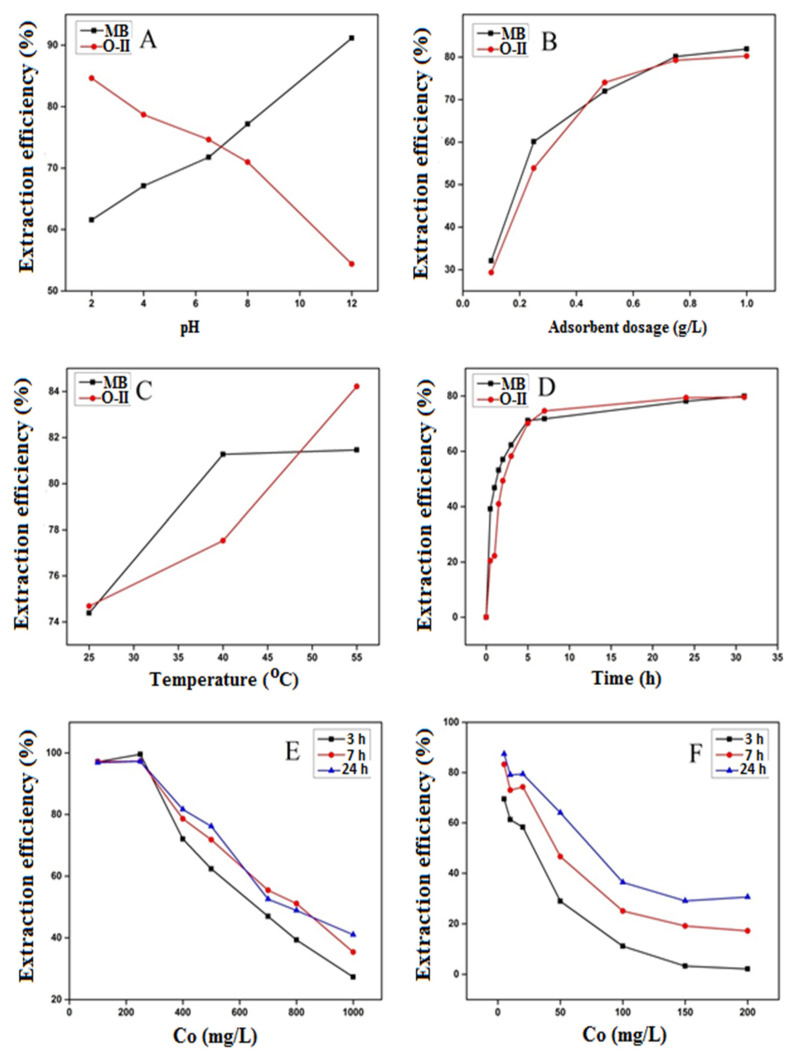
(**A**) Effect of pH (MB = 500 mg/L, O-II = 20 mg/L, m = 0.5 g/L, and t = 7 h), (**B**) adsorbent dosage (MB = 500 mg/L, O-II = 20 mg/L, pH = 7.0, and t = 7 h), (**C**) temperature (MB = 500 mg/L, O-II = 20 mg/L, m = 0.75 g/L, and t = 7 h), (**D**) contact time (MB = 500 mg/L, O-II = 20 mg/L, m = 0.75 g/L), (**E**) dye concentration (MB = 100–1000 mg/L, m = 0.75 g/L), (**F**) (O-II = 5–200 mg/L, m = 0.75 g/L).

**Figure 8 molecules-27-07775-f008:**
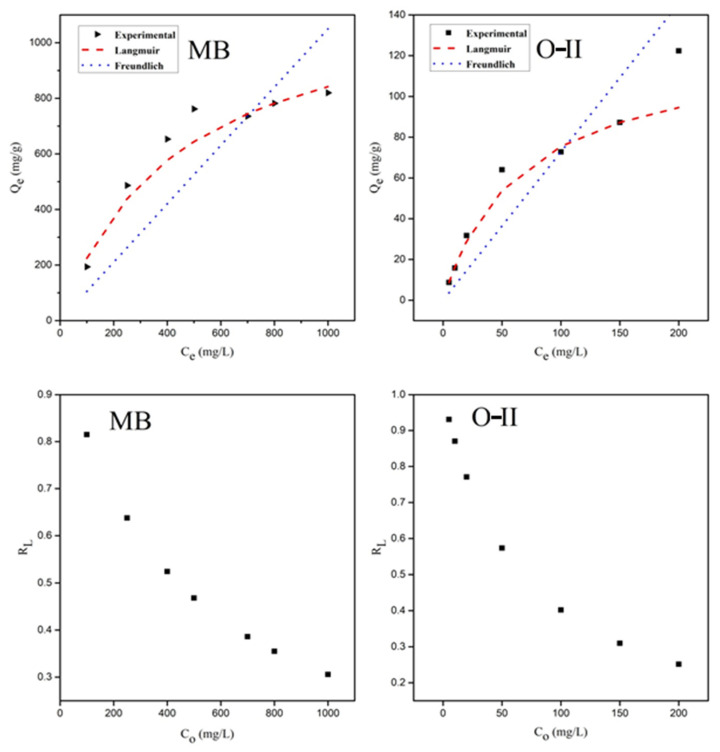
Langmuir isotherm model fitting and separation factor plots for MB and O-II.

**Figure 9 molecules-27-07775-f009:**
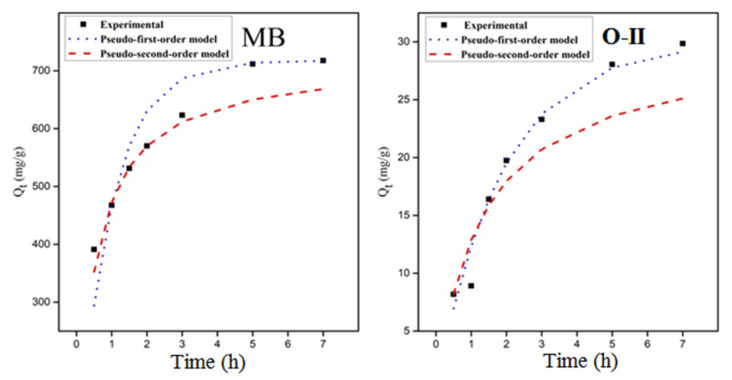
Kinetic plots for the adsorption efficiency of poly[veim][Tf_2_N]-TRIM towards MB and O-II dyes.

**Figure 10 molecules-27-07775-f010:**
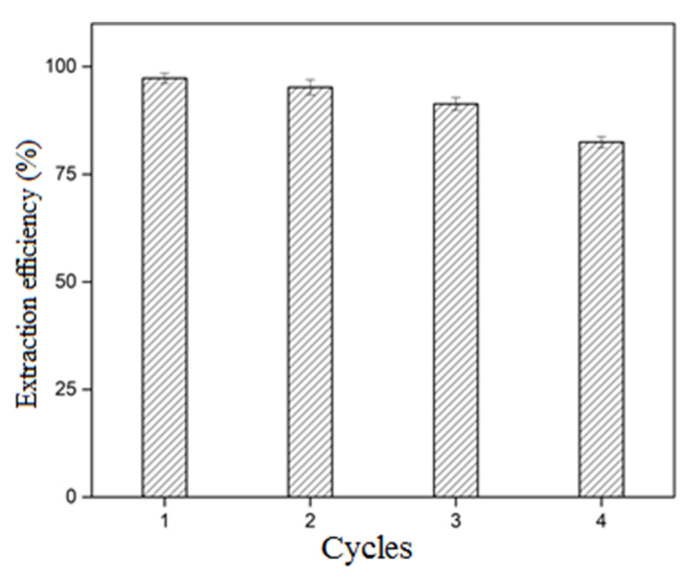
Extraction efficiency of poly[veim][Tf_2_N]−TRIM towards MB at 500 mg/L initial concentration in different cycles.

**Figure 11 molecules-27-07775-f011:**
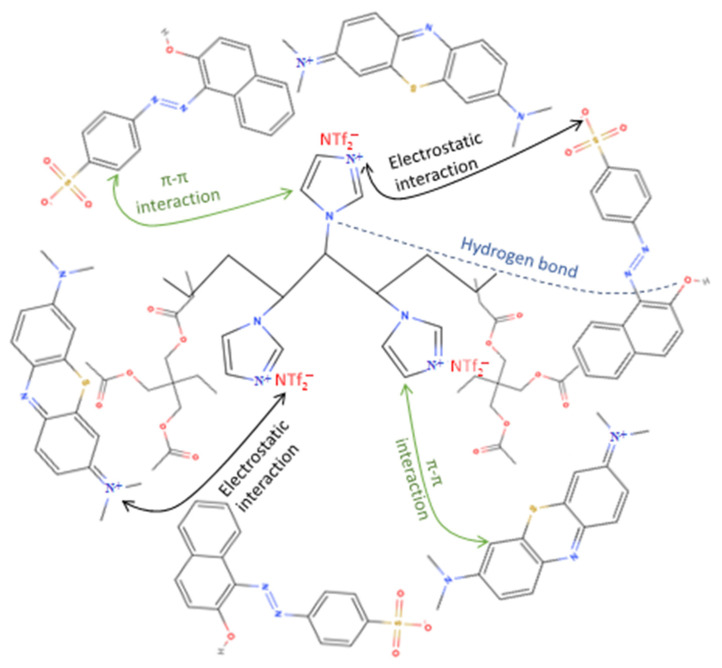
Schematic representation for the possible adsorption mechanism involved in this study.

**Table 1 molecules-27-07775-t001:** The kinetic parameters of MB and O-II dye adsorption.

Name of Dye	Experimental Adsorption	Pseudo-First-Order				Pseudo-Second-Order			
	Q_e,exp_	Q_e,cal_	k_1_	R^2^	ARE	Q_e,cal_	k_2_	R^2^	ARE
MB	717.7	717.3	1.054	0.951	0.076	668.3	0.0026	0.981	0.041
O-II	29.84	29.12	0.532	0.990	0.085	25.12	0.025	0.979	0.142

Q_e,exp_—experimental adsorption capacity; Q_e,cal_—calculated adsorption capacity.

**Table 2 molecules-27-07775-t002:** Adsorption isotherm parameters for MB and O-II dyes.

Dye Name	Langmuir Isotherm				Freundlich Isotherm			
	Q_m_ (mg/L)	K_L_	ARE	R^2^	K_F_	n	ARE	R^2^
MB	1212	0.002	0.081	0.968	10.26	9.77	0.277	0.870
O-II	126.32	0.014	0.078	0.970	7.97	10.95	0.362	0.960

**Table 3 molecules-27-07775-t003:** A relative comparison of adsorption efficiency of various adsorbents for the removal of MB and O-II dyes.

Name of Adsorbent	Name of Dye	Adsorption Capacity Q_m_ (mg/g)	Reference
**Poly[veim][Tf_2_N]−** **TRIM**	MB	1212	This study
Graphene hydrogel		660	[8]
Fe_3_O_4_–βCD–DCA polymer composite [Fe@CDA_2_]		333	[69]
Halloysite-Cyclodextrin nano sponges		226	[70]
PIL@PDA@Fe_3_O_4_		72	[24]
**Poly[veim][Tf_2_N]−** **TRIM**	O-II	126	This work
Apricot shell-AC		14	[71]
Amino-functionalized titanosilicate		189	[72]

## Data Availability

The data presented in this study are available on request from the corresponding author upon request.

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
