# Peer review of "Cross-Linked Ionic Liquid Polymer for the Effective Removal of Ionic Dyes from Aqueous Systems: Investigation of Kinetics and Adsorption Isotherms"

_molecules, 2022, doi:10.3390/molecules27227775_

Round 1

Reviewer 1 Report

The authors in the work reported the preparation of cross-linked ionic polymers for ionic dyes capture. I raised the following concerns.

1.       The design of these ionic polymers is not that reasonable. It’s well known that the polymers prepared from flexible cross-linker like TRIM usually form non-porous structures. The porous structures of the prepared ionic polymers are unclear according to the SEM images. Characterization of the textural properties, such as N2 sorption, is necessary.  

2.       As Figure 1 shows, only the ionic liquid monomer is surrounded by the initiator, and the initiator remains in the resulting poly[veim][Tf2N]-TRIM. This is absolutely wrong. More importantly, little information can be found in such as Figure 1.

3.       The authors discussed the adsorption performance in the characterization section. This is not well organized.

4.       According to the introduction section, the highlight of the work is the trapping efficiency of poly[veim][Tf2N]-TRIM to anionic and cationic dyes. Figure 6 presents the extraction efficiencies of ionic polymers with different anions toward both anionic and cationic dyes. Analysis of the results in Figure 6 is crucial, but it is not clear in the discussion section.

5.       The authors mentioned that poly[veim][Tf2N]-TRIM is efficient in capture of cationic dye. Analysis could show that the toxic portion of the dye was still in the solution but not in the heterogeneous polymer.

6.       The authors mentioned that there is a clear difference in surface morphology of poly[veim][Tf2N]-TRIM before and after dye adsorption. They should provide further evidence.

7.       The authors used choline chloride to regenerate the polymer after dye adsorption. That means the chemical structure of the regenerated material is different from the pristine material, at least they have different anions. So, how the regenerated material can retain the adsorption performance?

8.       The adsorption mechanism needs experimental and theoretical supports.

Reviewer 2 Report

Authors have used Ultrasonic functionalized cross-linked poly[veim][Tf2N]-TRIM for the adsorption of MB and Orange II dye. They have conducted adsorption characterization and batch adsorption studies. However, the following points to be considered. 

1. There are few typographical errors in the manuscript.

2. Authors need to report the point of zero charge of the adsorbent and need to explain the effect of solution pH on dye removal by using point of zero charge of the adsorbent.

3. Authors need to mention in detail the mechanism of dye adsorption onto adsorbent.

4. The desorption and regeneration study is missing the manuscript.

5. Dye containing wastewater is not a single component system hence, effect of other contaminants like surfactant, salinity on dye removal process by the adsorbent plays an important role. This study is missing in the manuscript.

6. Introduction: kindly include the scope and limitations of adsorbents for water treatment.

7. Essential related works can be cited. Authors can also include recent literature:

a) Industrial Crops and Products 153 (2020) 112613.
(https://doi.org/10.1016/j.indcrop.2020.112613)
b) International Journal of Environmental Research 13 (2019) 349-366.
(https://doi.org/10.1007/s41742-019-00181-0)
c) Science of The Total Environment, 812 (2022) 152456.
(https://doi.org/10.1016/j.scitotenv.2021.152456) 

8. Significant findings must be summarized in the abstract to make it more catching.

Reviewer 3 Report

Comments to authors

In this work, the author synthesized an imidazolium based cross-linked polymer called 1-vinyl-3-ethylimidazoliumbis(trifluoromethylsulfonyl)imide(poly[veim][Tf2N]-TRIM) and utilized for the removal of two extensively used ionic dyes namely methylene blue and orange-II from aqueous systems. The impact factors including initial dye concentration, contact time, adsorbent dosage, solution pH and temperature have been detailly assessed and optimized. Also, the adsorptive kinetics and mechanism of dyes adsorption have also been investigated and speculated in this work. All in all, I think this work is meaningful in the scientific research on environmental contaminants removal. However, there are some issues need to be addressed before accepted in this journal.

1. As all we known, the compositional ingredients of the practical effluents are complex, specially, salt is a major ingredient. Thus, the author should be assessed the influence of salt for example NaCl, Na2SO4, on the adsorptive capacity of dyes.

2. What role does ionic liquids in the polymers play in the dye adsorption and how does the amount of ionic liquid affect dye adsorption, the author should be described in the manuscript.

3. The author should be supplemented some recently developed novel polymer adsorbents in the introduction section, for instance, Journal of Environmental Chemical Engineering, 10(5), 108425.  Journal of Materials Chemistry A, 2021, 9(1): 254-258. etc.

4. There are some spelling and syntax mistakes. The language should be carefully polished.

Round 2

Reviewer 1 Report

The authors have improved their manuscript. However, my main concerns are still there.

The authors do not think the characterization of the textural properties is necessary. They have evaluated poly[veim][Tf2N] in their previous study, and think the addition of the cross-linker (TRIM) has no effect on the porosity of resulted polymer. However, cross-linker will normally affect resulted polymers.

The authors do not think that the toxic portion of the dye is in the solution. However, for the polymer with [eim]+[TF2N]- and MB with [A]+Cl- (A represents the complex cationic part of MB, the toxic part), [TF2N]- and the toxic part [A]+ are in the solution after ion exchange.

The authors still think that the regenerated material is the same to the pristine material. However, for the polymer with [eim]+[TF2N]- and choline chloride with [B]+Cl-(B represents the cationic part of choline chloride), the regenerated polymer material is a new one with [eim]+Cl- which is different from the original polymer material with [eim]+[TF2N]- after ion exchange.
